evolution

wolf, *Canis lupus*, whole genome, demographic history, Iberian wolf, Italian wolf

**Author for correspondence:**
Pedro Silva
e-mail: pedrosilva@cibio.up.pt

# Genomic evidence for the Old divergence of Southern European wolf populations

Pedro Silva[1], Marco Galaverni[2], Diego Ortega-Del Vecchyo[3], Zhenxin Fan[4], Romolo Caniglia[5], Elena Fabbri[5], Ettore Randi[6,7], Robert Wayne[8] and Raquel Godinho[1,9,10]

[1]CIBIO/InBIO – Centro de Investigação em Biodiversidade e Recursos Genéticos, Universidade do Porto, Campus Agrário de Vairão, 4485-661 Vairão, Portugal
[2]Conservation Unit, WWF Italia, Via Po 25/c - 00198 Roma, Italy
[3]International Laboratory for Human Genome Research, National Autonomous University of Mexico, Santiago de Querétaro, Querétaro 76230, Mexico
[4]Key Laboratory of Bioresources and Ecoenvironment (Ministry of Education), College of Life Sciences, Sichuan University, Chengdu, People's Republic of China
[5]Unit for Conservation Genetics (BIO-CGE), Department for the Monitoring and Protection of the Environment and for Biodiversity Conservation, Italian Institute for Environmental Protection and Research (ISPRA), Via Cà Fornacetta 9, 40064 Ozzano dell'Emilia (Bo), Italy
[6]Department of Biological, Geological and Environmental Sciences, University of Bologna, Via Selmi 3, Bologna 40126, Italy
[7]Department of Chemistry and Bioscience, Faculty of Engineering and Science, University of Aalborg, Aalborg, Denmark
[8]Department of Ecology and Evolutionary Biology, University of California, Los Angeles, CA, USA
[9]Departamento de Biologia, Faculdade de Ciências, Universidade do Porto, 4169-007 Porto, Portugal
[10]Department of Zoology, Faculty of Sciences, University of Johannesburg, Auckland Park 2006, South Africa

PS, 0000-0002-0362-7840; RG, 0000-0002-4260-4799

The grey wolf (*Canis lupus*) is one of the most widely distributed mammals in which a variety of distinct populations have been described. However, given their currently fragmented distribution and recent history of human-induced population decline, little is known about the events that led to their differentiation. Based on the analysis of whole canid genomes, we examined the divergence times between Southern European wolf populations and their ancient demographic history. We found that all present-day Eurasian wolves share a common ancestor *ca* 36 000 years ago, supporting the hypothesis that all extant wolves derive from a single population that subsequently expanded after the Last Glacial Maximum. We also estimated that the currently isolated European populations of the Iberian Peninsula, Italy and the Dinarics-Balkans diverged very closely in time, *ca* 10 500 years ago, and maintained negligible gene flow ever since. This indicates that the current genetic and morphological distinctiveness of Iberian and Italian wolves can be attributed to their isolation dating back to the end of the Pleistocene, predating the recent human-induced extinction of wolves in Central Europe by several millennia.

## 1. Introduction

Grey wolves (*Canis lupus*) were once widely distributed as top predators in many ecosystems across the Holarctic [1]. Grey wolves first arrived in Europe at the end of the Middle Pleistocene, *ca* 0.3–0.5 million years ago (Mya) [2], and have lived through the profound environmental changes of the cyclical glaciations and the transition to the Holocene that greatly affected patterns of genetic diversity and differentiation of many species in the Northern hemisphere [3,4]. The fossil record suggests that wolves have maintained a large distribution during this period, with no evidence of retraction to glacial refugia [2,5]. The first phylogeographic studies based on mitochondrial DNA (mtDNA)

found relatively recent coalescent times and no large-scale geographic structure, which was hypothesized to be the result of repeated population contractions and rapid recolonizations during glacial cycles [6]. However, the scenario emerging from several recent studies based on whole mtDNA and nuclear genomic data suggests that all extant wolves in the New and Old World derive from a single ancestral population that expanded worldwide during the last glaciations [7–12]. Therefore, despite their much older age, it appears that the defining genetic history of modern wolves has occurred over the last 30 000 years [9,11,12].

In the last few centuries, reductions in available habitat and natural prey, as well as direct human persecution, have resulted in the extinction of wolves in most of Central and Western Europe, and parts of North America [13]. Remaining populations became greatly reduced and fragmented, following the same trend as other large carnivores [14,15]. In recent decades, due to the implementation of protection measures and changes in land use, some wolf populations have recovered and successfully reinvaded areas where they were previously extirpated, aided by their ability to disperse rapidly and over long distances [15]. Although still suffering significant reductions and fragmentation, Eastern European wolves, such as the Dinaric-Balkan and Carpathian populations, remained relatively larger and interconnected, and in contact with populations from Russia and the rest of Asia [16]. By contrast, the Southern European wolf populations of the Iberia and Italy represent unique relics from before the mass extirpation, as they became effectively isolated in their respective peninsulas since the end of the 19th or beginning of the 20th centuries [14,17,18].

Italian and Iberian wolves are consistently reported as having low genetic diversity and being highly differentiated when compared to other European or most worldwide populations (e.g. [6,9,19–22]). This pattern can be partially explained by the severe decline that both populations experienced during the last two centuries. Specifically, they reached their minimum in the 1970s, when less than 100 wolves survived in Italy [18] and 500 to 700 wolves are estimated to have remained in the Iberian Peninsula [23,24]. The distinctiveness of these wolves is also reflected in their morphology, given that they generally have a smaller body size, a typical fur appearance and can be differentiated from other wolves on several skull measures [25–27]. This morphologic distinctiveness was the basis for the description of two subspecies in the beginning of the twentieth century: *C. l. signatus* [28] and *C. l. italicus* [29] for the Iberian and Italian wolves, respectively. These classifications, while recently recognized in the IUCN Red List [30], have not been widely accepted (see, for example, [31,32]). The mentioned genetic and morphologic distinctiveness of these wolves has also been taken as an indication that these populations may have been isolated for far longer than the last couple of centuries. Genetic studies, based on genome-wide microsatellite or SNP markers have indeed supported this hypothesis, providing, however, a wide range of divergence times from 2000–19 000 years ago (kya) [19], 3–6 kya [21] and 20 kya [33].

In this study, we leverage the availability of full genome data from canids worldwide [8,9] to further investigate the demographic history of Southern European wolves, focusing specifically on the Iberian and Italian populations, including the timing of their divergence, levels of gene flow and long-term effective population sizes. For this, we used a demographic inference method capable of integrating information from many unlinked genomic segments [34]. In particular, we attempt to clarify if these populations have been isolated for some centuries (possibly due to the aforementioned human impacts and extermination) or for a far longer time. Understanding this specific time frame has implications in terms of phylogeography, taxonomy and conservation of these populations. Regarding phylogeography, many species of plants and animals in Europe show a classic phylogeographic pattern of differentiation resulting from the contraction to glacial refugia in the Southern peninsulas and posterior expansion during the cyclical Pleistocene glaciations [3,4,35]. It is therefore intriguing that wolves of the Iberian and Italian peninsulas show a morphological and genetic differentiation without an apparent range contraction during the glaciations [2,5]. Furthermore, better divergence time estimates could help elucidate their inconsistent taxonomic classification [19]. From a conservation perspective, and given that these wolves also represent some of the last surviving large predators in Southern Europe, a better understanding of their evolutionary history allows a better assessment of their expected levels of genetic diversity, an important parameter in designing management and conservation measures. We interpret our findings in the light of recorded population declines in recent centuries as well as older climatic conditions of the Pleistocene to understand the influence ancient and recent events may have had in shaping the genetic composition of these wolf populations.

## 2. Material and methods

### (a) Canid genome data

We compiled a dataset of full genome sequences from nine canids at 12–26 × average coverage (table 1), in addition to the canFam3.1 dog reference genome [36]. Six canids have been sequenced by [8] and include wolves from Croatia, Israel and China, dogs of the dingo and basenji breeds, and one golden jackal (*Canis aureus*) from Israel that was used as an outgroup. In addition, we included three European wolves from [9]: two individuals from the Iberian Peninsula (Portuguese and Spanish wolves) and one from Italy. We used the final genotype calls in Variant Call Format (VCF) from the mentioned studies, without any additional data processing. Details regarding read alignment, genotyping and quality filtering procedures can be found in the original studies, namely in electronic supplementary material, texts 3 and 4 of [8] and the electronic supplementary material, text of [9]. The original sequencing data are available at the NCBI Sequence Read Archive (SRA; https://www.ncbi.nlm.nih.gov/sra/) with accession numbers SRP062184 and SRP044399. The geographic origin of the samples in relation to the current wolf distribution in Eurasia is represented in figure 1.

### (b) Estimating population divergence times, effective sizes and migration

We performed demographic inferences on the individual genomic sequence data using the Generalized Phylogenetic Coalescent Sampler (G-PhoCS 1.3 [34]). For a given population phylogeny, G-PhoCS infers effective population sizes, population divergence times and migration rates between population pairs likely to have experienced post-divergence gene flow. These demographic parameters are estimated based on inferred genealogies for thousands of neutrally evolving loci; therefore, a set of

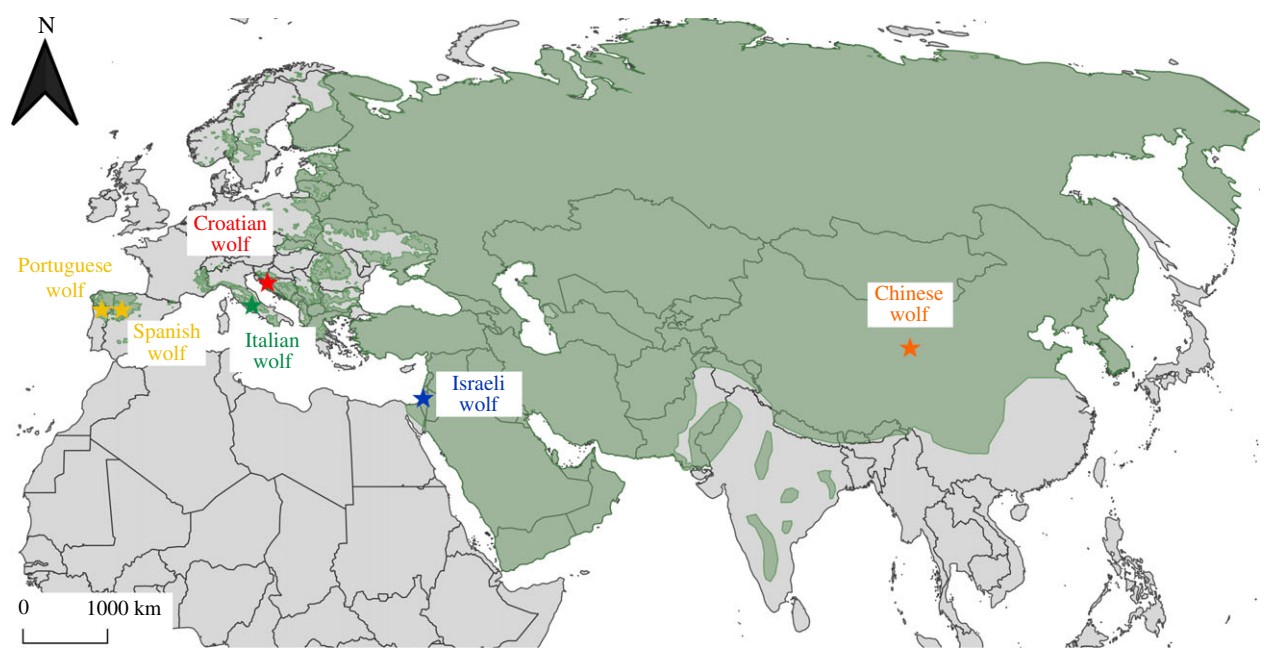

**Figure 1.** Map with the location of grey wolf samples used in the G-PhoCS analysis. Colours of individual wolves correspond to their respective populations, as referred in the main text: yellow: Iberian; green: Italian; red: Dinaric-Balkan; blue: Middle Eastern; orange: Chinese. Current wolf range is represented in green, according to the IUCN Red List [30]. (Online version in colour.)

**Table 1.** Canid genomes used in this study in addition to the dog (boxer) canFam3.1 reference genome.

| sample | region of origin | average genome coverage (X) | reference |
|---|---|---|---|
| Portuguese wolf | Minho, Portugal | 26.1 | [9] |
| Spanish wolf | Castilla y Léon, Spain | 25.3 | [9] |
| Italian wolf | Calabria, Italy | 13 | [9] |
| Croatian wolf | Perković, Croatia | 25.3 | [8] |
| Israeli wolf | Neve Ativ, Golan Heights, Israel | 21.6 | [8] |
| Chinese wolf | San Diego Zoo, California, USA | 24.6 | [8] |
| basenji | Maryland, USA | 12.6 | [8] |
| dingo | Bargo Dingo Sanctuary, Australia | 25.8 | [8] |
| golden jackal | Tel Aviv, Israel | 23.8 | [8] |

high-quality genomic regions that ideally have not been subject to strong selection are needed [34]. We used the same putatively neutral regions defined by [8] (electronic supplementary material, text 9.2.1 therein). Briefly, we excluded regions of the genome with assembly gaps, repeats, low mappability, missing data in all samples (i.e. bases not passing quality filters), coding regions (and respective 10 kb flanking regions) and regions that are highly conserved in mammals. In total, 13 696 1 kb regions of the genome remained after applying these filters. For these regions, we constructed multiple sequence alignments for our samples in addition to the boxer reference genome (canFam3.1, [36]). Individual positions failing quality filters were masked as N's; additionally, all 'CG' dinucleotides, as well as all positions with a C* dinucleotide in one genome and *G in another were also masked.

We assumed a population tree according to the maximum-likelihood phylogeny inferred by [9] (figure 2). MCMC runs were executed using the same settings as in [34] and [8] (electronic supplementary material, text 9.2.2 of [8]; electronic supplementary material, table S1). The program was allowed to run for 500 000 iterations, the first 200 000 of which were discarded as burn-in. The average runtime for this full analysis

was approximately 8 days using 16 threads and 16 GB of RAM on a compute node equipped with AMD Opteron Processor 6380 CPUs. Convergence was inspected manually for each run using Tracer 1.7 [37].

Given the limitations of the method when a large number of migration bands are used, we followed the same procedure as [8] (electronic supplementary material, text 9.3 therein) and used a two-step approach to infer gene flow in our data. First, we identified which population pairs showed signs of significant gene flow in two separate runs using the same settings as in the final run but using only a subset of 5000 of the loci. In one run, we considered migration bands between geographically adjacent wolf populations (Portuguese wolf ↔ Spanish wolf, ancestral Iberian wolves ↔ ancestral Italian wolves, Italian wolf ↔ Croatian wolf, Croatian wolf ↔ Israeli wolf, Israeli wolf ↔ Chinese wolf) and in the other run, between every wolf population and the dogs. A migration band was inferred to have significant gene flow if the 95% highest posterior density interval (HPDI) of the total migration rate for that band did not include 0, or if the total migration rate was estimated to be greater than 0.03 with posterior probability greater than 50%. We then performed a final run with

migration bands between the pairs that showed significant gene flow.

Parameters estimates $\theta$ (population size) and $\tau$ (divergence time) given by G-PhoCS are scaled by the mutation rate and generation time. Effective population sizes in number of individuals ($N_e$) can be calculated by $\theta = 4N_e\mu$, and divergence times in years (T) by $\tau = T\mu/g$, where $\mu$ is the mutation rate (per base pair, per generation) and g is the average generation time (in years). Given the uncertainty in wolf/dog mutation rates and generation times (see Discussion), we consider two alternative combinations of these values for converting our results: (1) $4 \times 10^{-9}$ mutations/bp/generation and 4.5 years/generation; (2) $1 \times 10^{-8}$ mutations/bp/generation and 3 years/generation.

The G-PhoCS model also uses a scaled version of migration rate, $M = m/\mu$, where m is the probability of migration between two populations in a single generation. The level of gene flow across a given migration band is measured by the total migration rate, which is the migration rate scaled by the time span of the migration band ($\tau_m$): $m_{tot} = M\tau_m$. The time span of a migration band is defined using the start and end times of the two populations involved. By scaling the migration rate M by the time span $\tau_m$, total migration rates $m_{tot}$ are independent of the assumptions regarding mutation rate.

## (c) Evaluation of generalized phylogenetic coalescent sampler results using simulations

To confirm the reliability of the main analysis, we reran G-PhoCS with simulated data. For all the simulations, we assumed a recombination rate of $0.92 \times 10^{-8}$/bp/generation, based on the mean recombination rate estimated from a linkage map of the domestic dog genome, constructed from microsatellite data [38]. The program ms [39] was used to simulate gene trees at 15 000 loci under the parameters inferred by G-PhoCS in the analyses described above (command line 1 in the electronic supplementary material). Seq-gen [40] was then used to build 1 kb sequences for each of those loci (command line 2 in the electronic supplementary material). We assumed the simulated loci evolved under the JC69 model [41]. The alignments produced in this manner were then used as input for G-PhoCS using the same settings as previously described for the main analyses. In our control simulation, all parameters corresponded to the ones inferred by G-PhoCS in the analysis described above. Given the unexpected high divergence time between the Portuguese and Spanish wolves (see Results), we performed two additional simulations to verify that G-PhoCS is capable of detecting very recent split times. In one of these simulations, the divergence time between the Portuguese and the Spanish wolves was assumed to be 90% smaller than inferred in the main analysis (command line 3 in the electronic supplementary material), and in the other, complete gene flow (full panmixia) exists until the present between the Portuguese and Spanish populations (command line 4 in the electronic supplementary material).

## (d) Robustness of generalized phylogenetic coalescent sampler results to dog ancestry

Given the significant gene flow between dogs and the Spanish wolf, inferred in our main G-PhoCS analysis (see Results), as well as the significant dog ancestry displayed by this sample in other studies [9,42], we tested the impact of this shared ancestry in our results. We applied the *Admixture* tool (v1.2.0) from the *Galaxy* platform (www.usegalaxy.org, [43]) to identify regions of the Spanish wolf genome displaying significant dog ancestry. We considered the three dog genomes (boxer, basenji and dingo) as representing one source population and used the Croatian and Chinese wolves as the other source population, since the Israeli sample also shows traces of admixture with dogs [9]. We only retained sites that presented an $F_{ST}$ value $\geq 0.5$ between the two source populations (572130 sites) and used a penalty switch value of 50. We then repeated the G-PhoCS analysis removing any 1 kb sequence not contained within the genomic regions identified as wolf by Galaxy/Admixture (8553 of the 13 696 regions were removed). Additionally, to assess the impact of removing such a high proportion of sequences, we repeated the analysis removing a similar number of random regions. Finally, we also repeated the whole G-PhoCS analysis removing the Spanish wolf sample entirely.

## 3. Results

Divergence times, long-term effective population sizes and migration rates for European wolf populations were estimated using whole genome sequencing data and two alternative combinations of mutation rates and generation times: $4 \times 10^{-9}$ mutations/bp/generation and 4.5 years/generation, or $1 \times 10^{-8}$ mutations/bp/generation and 3 years/generation (figure 2 and electronic supplementary material, tables S2 and S3). We report in this section the highest and lowest estimates of divergence time and effective population size as obtained by using these two combinations. Each point estimate is followed by the respective 95% HPDI.

The wolf/dog divergence is dated at *ca* 36 kya (33 480–38 858) or, alternatively, 9.6 kya (8928–10 362) ($T_{ancDW}$, electronic supplementary material, table S2). The divergence of East Asian and Middle Eastern wolf populations, represented in this analysis by the Chinese and Israeli wolves, appears to have been established shortly thereafter, *ca* 34 kya or 9 kya ($T_{ancWOLF}$: 31 016–36 338/8271–9690; $T_{ancIS-CR-IT-IB}$: 30 713–36 304/8190–9681, electronic supplementary material, table S2). The split between the sampled European wolf populations (Croatian, Italian and Iberian) occurred much more recently and very close together, with estimates between *ca* 11 kya or 3 kya: the Croatian wolves diverged from the others *ca* 10.8 kya (7920–12 668) or 3 kya (2112–3378)($T_{ancCR-IT-IB}$, electronic supplementary material, table S2), and Italian wolves diverged from Iberian wolves *ca* 10.3 kya (7211–12 229) or 3 kya (1923–3261)($T_{ancIT-IB}$, electronic supplementary material, table S2). The two samples from the Iberian population (Portuguese and Spanish wolves) are estimated to have diverged from each other *ca* 6 kya (3071–8753), or 2 kya (819–2334)($T_{ancIB}$, electronic supplementary material. table S2). Divergence time estimates based on simulated data with (i) a lower divergence time or (ii) panmixia between these two wolf populations show that G-PhoCS would accurately infer lower split times in those two scenarios ($T_{ancIB}$, electronic supplementary material, table S4). In other words, G-PhoCS appears to provide accurate divergence time inferences from recently very diverged genomes, adding further confidence that the obtained old divergence estimate between Portuguese and Spanish wolves results from the actual information in the data.

The ancestral population from which all wolves and dogs sampled in this study originate ($N_{ancDW}$, electronic supplementary material, table S2) is estimated to have had a much larger effective size than either of the descendent populations ($N_{ancWOLF}$ and $N_{ancDOG}$, electronic supplementary material, table S2). Specifically, we infer $8\times$ and $10\times$ reductions for the populations ancestral to dogs and

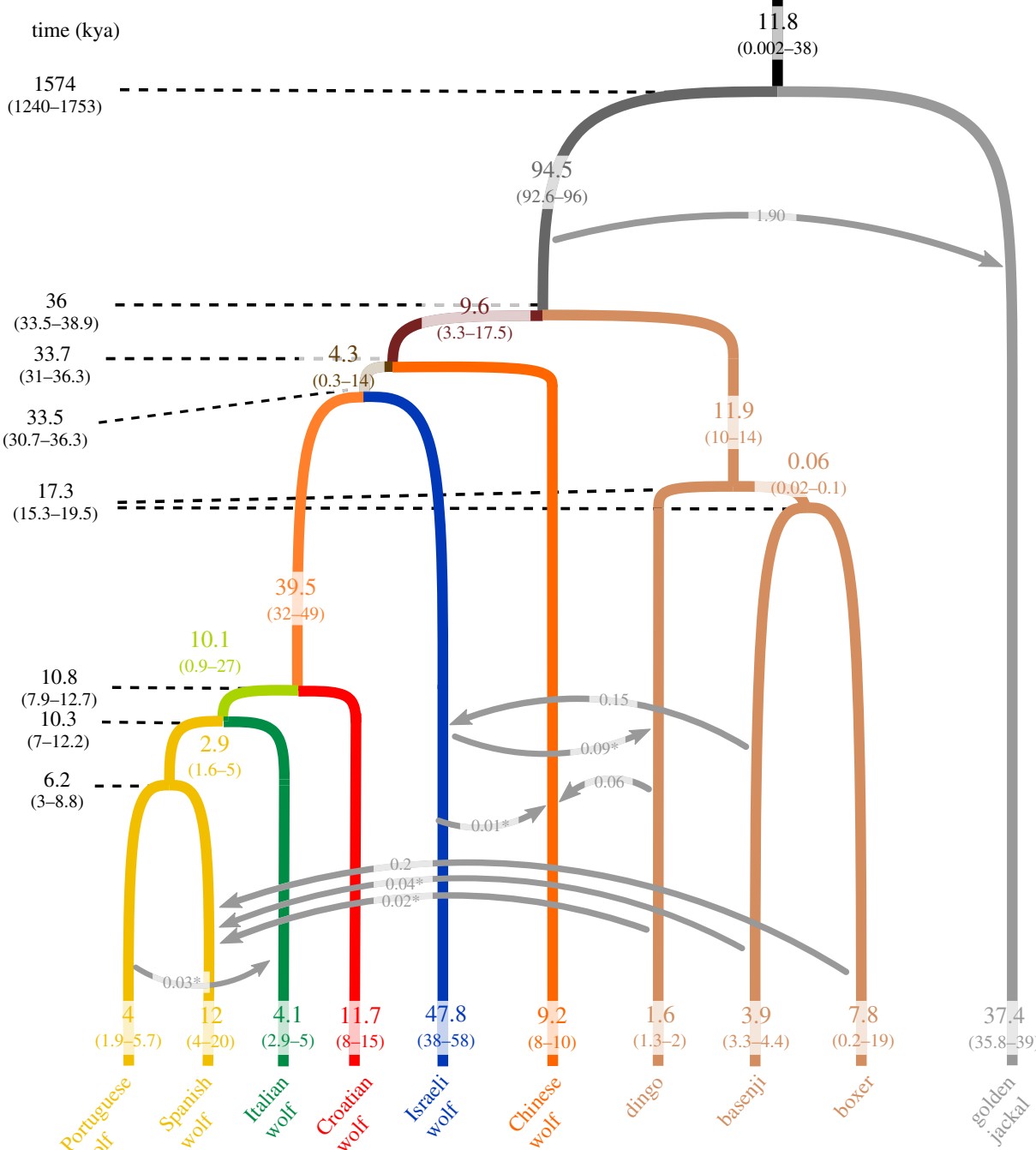

**Figure 2.** Demographic model and parameter estimates tested in G-PhoCS. Branch colours of current populations are the same as in figure 1. Values on branches are inferred effective population sizes in thousands of individuals; arrows represent migration; values to the left of the diagram are divergence times in thousands of years. Values were converted using a mutation rate of $4 \times 10^{-9}$/bp/generation and a generation time of 4.5 years. For each parameter, mean values and 95% HPDI are shown. Migration rates (on the arrows) with an asterisk are estimates whose 95% HPDI include 0. (Online version in colour.)

wolves, respectively. Depending on the assumed mutation rate, the ancestral population had an $N_e$ of the order of 94 000 individuals (92 586–96 436), or 38 000 individuals (37 034–38 574)($N_{ancDW}$, electronic supplementary material, table S2). Ancestral dog $N_e$ estimates are *ca* 12 000 individuals (10 234–13 813) or 4800 individuals (4094–5525)($N_{ancDOG}$, electronic supplementary material, table S1), while ancestral wolf $N_e$ are *ca* 9600 (3228–17 458), or 3900 (1291–6983)($N_{ancWOLF}$, electronic supplementary material, table S2).

The population ancestral to sampled European wolves (Croatian, Italian and Iberian), at *ca* 40 000 individuals (31 554–48 599), or 16 000 individuals (12 622–19 440) ($N_{ancCR-IT-IB}$, electronic supplementary material, table S2), is

inferred to have been 10× larger than its ancestral wolf population ($N_{IS-CR-IT-IB}$, electronic supplementary material, table S2). However, Italian and Iberian wolves are inferred to have descended from populations that were much smaller. The population ancestral to Italian and Iberian wolves is estimated to have numbered *ca* 10 000 individuals (848–26 774), or *ca* 4000 individuals (339–10 710)($N_{ancIT-IB}$, electronic supplementary material. table S2), which represents a fourfold reduction in relation to the ancestral European wolf population. Estimates for the ancestral Iberian population are *ca* 3000 individuals (1637–5040), or *ca* 1000 individuals (655–2016)($N_{ancIB}$, electronic supplementary material, table S2). Since their divergence from European wolves, Middle

Eastern populations, here represented by the Israeli wolf, have maintained large population sizes in the order of almost 50 000 (38 361–57 740) or 20 000 individuals (15 344–23 096)($N_{ISW}$, electronic supplementary material, table S2). Within individual European wolf populations, the smallest effective sizes are inferred for the Italian wolf ($N_{ITW}$, electronic supplementary material, table S2: 2887–5104 or 1155–2042) and the Portuguese wolf ($N_{PTW}$, electronic supplementary material, table S2: 1890–5656 or 756–2262). Sizes inferred for the Spanish and Croatian wolves are three times higher.

Regarding post-divergence gene exchange, we detected significant gene flow from basenji to Israeli wolf (approx. 15%), from the dingo to the Chinese wolf (approx. 6%) and from the boxer to the Spanish wolf (approx. 20%)(electronic supplementary material, table S3). Very low levels of gene flow, whose 95% HPD interval includes 0, were inferred from the Israeli wolf to the dingo (9%) and to the Chinese wolf (approx. 1%), from the Portuguese wolf to the Italian wolf (approx. 3%), from the dingo to the Chinese wolf (approx. 6%), and from the basenji (approx. 4%) and the dingo (approx. 2%) to the Spanish wolf (electronic supplementary material, table S3).

G-PhoCS divergence times were not affected by the dog ancestry in the Spanish wolf (electronic supplementary material, table S5). Removing genomic regions with dog ancestry from the Spanish wolf sample leads to generally similar parameter estimates, albeit with some differences regarding some effective population sizes. Specifically, the size of the Spanish wolf population is similar to the Portuguese wolf when dog regions are removed ($N_{SPW}$, electronic supplementary material, table S5). This effect is maintained if a similar number of regions is removed at random from the Spanish wolf but not if this sample is omitted entirely, in which case results are similar to the ones of the main analysis. This indicates that the differences in the size estimate for the wolf ancestral population is due to the inclusion of a sample with fewer usable loci and not due to dog ancestry.

## 4. Discussion

Our demographic analyses of whole genomes confirm that the history of Eurasian wolf populations has been shaped by significant population fluctuations and divisions. Even taking into account the uncertainty regarding the timing of these events, they appear to be older than the fragmentation and decline that occurred in recent centuries by human persecution and habitat changes.

The exact divergence time and effective population size estimates are heavily dependent on the values assumed for the mutation rate and wolf generation time. Considerable uncertainty exists about both of these figures in the literature. Regarding generation time, most genetic studies focused on wolves and dogs, mainly in the context of ascertaining the time of dog domestication, assume a value of 3 years, e.g. [8,9,21,33,36,44–47]. It is unclear, however, on what information this value is based and, as pointed out by [48], this value may be derived from a report on a single individual [49]. Studies performed on North American wolves with large sample sizes and employing different techniques estimated generation times of 4.16 [50] and 4.3–4.7 years [51]. By contrast, disparate values have been used for the nuclear

mutation rate. A rate of $1 \times 10^{-8}$ mutations/bp/generation (excluding CG substitutions), similar to the human mutation rate, has been assumed as a reasonable value in some studies, e.g. [8,9,36,44,52]. Nuclear sequence comparisons between mammal lineages have yielded genome-wide estimates (i.e. including all types of substitutions) of $1.8 \times 10^{-9}$ to $2.2 \times 10^{-9}$ substitutions/bp/year [47,53,54], which would correspond to per generation rates of $5.4 \times 10^{-9}$ to $9.9 \times 10^{-9}$, depending on the generation time assumed. Studies that used genomic sequences from ancient wolf and dog samples have estimated even lower mutation rates of $4 \times 10^{-9}$ [45] and $3–4.5 \times 10^{-9}$ mutations/bp/generation [46] (both excluding CG substitutions). A recent study using wolf parent–offspring trios provides concordant rates, with a point estimate of $4.5 \times 10^{-9}$ ($2.6–7.1 \times 10^{-9}$) [55]. In this work, we considered two combinations of mutation rate and generation time: (i) $4 \times 10^{-9}$ mutations/bp/generation and 4.5 years/generation and (ii) $1 \times 10^{-8}$ mutations/bp/generation and 3 years/generation. The first combination leads to older divergence times and higher effective population sizes relative to the second. We discuss our results using combination (i), given that recent evidence seems to support these values, but also present values converted using combination (ii) as a comparison with previous studies.

Independently of the parameter conversions used, our demographic results provide evidence for two important events in the history of Eurasian wolves, and of the Southern European populations in particular. First, our model supports the hypothesis that all modern Eurasian wolves descend from a small ancestral population which represents only a fraction of the genetic diversity of ancient wolves. Second, the isolation of Southern European wolf populations appears to predate the extirpation of Central European populations by several millennia, possibly dating back to the end of the Pleistocene.

The drastic bottleneck of ancient wolves is apparent in our results when comparing the effective population sizes of the population ancestral to all wolves and dogs and that ancestral to wolves (ancDW and ancWOLF in electronic supplementary material, table S1, respectively): the latter represents *ca* one-tenth of the former. This result supports the idea of profound changes in wolf populations during the climatically unstable glacial periods. Although a decrease in genetic diversity during the late Pleistocene has been inferred based on different genetic data (e.g. [6,8,9,12,20]), no decrease in wolf distribution, compatible with a retreat into glacial refugia in Europe is apparent in the fossil record [5]. In fact, contrary to other predators that went extinct in Eurasia around the transition to the Holocene (e.g. cave bears, lions and hyenas), wolves survived and maintained a wide geographical range throughout this period [2]. Therefore, the pattern we observe likely results from changes in the relative abundance of different wolf lineages, in accordance with a recent hypothesis that all modern wolves descend from a single lineage which only expanded to its full worldwide distribution around the last glaciation, possibly replacing other wolf forms adapted to different conditions [8,9,11]. Comparisons of morphology [56] and isotope composition [57] between late Pleistocene and Holocene wolves indicate significant dietary differences, suggesting significant changes in their ecological role. The extinction of a locally adapted wolf ecomorph and replacement by a more generalized form during the later

Pleistocene has been described in North America based on ancient and modern genetic samples [7,10]. A recent study of ancient and modern wolf samples suggests that this modern lineage originated in Beringia and expanded over the Northern Hemisphere at the end of the Last Glacial Maximum (LGM), ca 25 kya [11]. This hypothesis has also been proposed as an explanation for the difficulty in determining the geographic origin of domestic dogs, as these could derive from a now extinct wolf population, implying that no extant wolf population is genetically closest to dogs [8,9,11,58].

We estimate that the divergence of the modern Eurasian wolf population, and the consequent bottleneck relative to its ancestral population, occurred as long ago as approximately 36 (95% HPDI: 39–33.5) kya. This value is similar to previous estimates by [9] (32–28 kya, which would correspond to 48–42 kya if a generation time of 4.5 years is used) which also included some of the same full genome sequences (Croatian, Israeli and Chinese wolves), and those of [11], who find a most recent common ancestor of extant Eurasian and American mitochondrial genomes ca 40 kya. This date follows the entrance and expansion of anatomically modern humans in Eurasia, ca 40–45 kya [59]. Therefore, environmental changes of the late Pleistocene could have allowed the expansion of humans concurrently led to the replacement of wolf forms, and/or that humans themselves might have had an impact on wolf populations [60]. Abrupt climate changes during the warmer interglacial periods (i.e. before the LGM approximately 23–19 kya) have been associated with the extinction or replacement of several megafaunal species [61], as well as human expansion in Europe [62]. Our results also imply that the domestication of dogs occurred near the same time as the wolf bottleneck, a dating that is consistent with other genomic studies [8,9,47,52] and the existence of dog-like remains greater than 30 000 years old [63,64]. Parameter conversion using the alternative combination of mutation rate and generation time would place the population size reduction and the domestication event at ca 9.7–10 kya. This date would correspond to the end of the last glacial period but would be incongruent with the genetic and fossil evidence mentioned above.

Our second main inference is related to the genetic isolation of the current Southern European wolves, which appears to be much older than the extirpation of Central European populations. Our analyses suggest that the divergence of the populations from Iberia, Italy and the Dinaric-Balkans occurred very closely in time, followed by negligible gene flow between them (all migration estimates 95% HPDIs include 0). 'Founding bottlenecks' for the Italian and Iberian populations seem to have been considerable and, although the Iberian population currently appears to have a higher effective population size than the respective founding population, Italian wolves appear to have declined further. The split between the three populations is estimated to have occurred ca 10.5 (95% HPDI: 7.2–12.7) kya and would therefore follow the end of the last glacial period (ca 11.7 kya), which was characterized by an abrupt but temporary return to glacial conditions. Another wave of megafauna extinctions took place around this time [61], suggesting profound environmental changes and/or human impacts that could also have affected European wolves.

Our results support the long-term isolation of Italian and Iberian wolf populations, which may explain the genetic and morphological distinctiveness exhibited by these wolves. While the divergence of the Croatian wolf from the Italian and Iberian populations is of the same magnitude, the absence of a sample from Eastern Europe or European Russia precludes us from inferring the degree by which the Dinaric-Balkan population has been isolated from these nearest populations. In our sampling, the nearest outgroup to the European clade is the Israeli wolf, which diverged ca 33.5 kya and for which no gene flow with European wolves is inferred. The comparatively higher effective population size estimated for the Croatian population might be indicative of less severe human impacts in the last few centuries or a higher connectivity with neighbouring Eastern populations.

Ancient divergence times for the European populations have been estimated before, although different studies proposed widely different estimates. An ancient isolation of the Italian wolf population was first proposed based on microsatellite data: modelling the demographic history as a continuous decline, the onset of population size reduction was estimated ca 2–4 kya, or as long ago as 19 kya, depending on assumptions of effective population size estimate and sampling strategy [19]. A more recent study, also based on microsatellite data, modelled the divergence of the three southern European wolf populations (Dinaric-Balkans, Italian and Iberian) as a simultaneous split that occurred 20 kya, which would correspond to 31 kya with a generation time of 4.5 years [33]. Divergence times estimated from genome-wide SNP data have a smaller uncertainty and are more in accordance with our results: the divergence between the Iberian and Italian populations has been estimated at 3.2–5.6 kya (4.8–8.4 kya if a generation time of 4.5 years is assumed) [21]. However, these values were considered as probably underestimated because of the calibration point used [21]. Our estimates, based for the first time on whole genome data, fit very well and with relatively narrow uncertainty with these previous results.

As with many other species, current Southern European wolves might represent populations that were mostly isolated from each other in their respective peninsular refugia during the Pleistocene glaciations. Reconnection of these populations after the return to warmer conditions might have been hindered by unknown ecological factors or even early human influence. However, there is also no evidence that the intervening wolf populations in Central Europe were absent during this period [5], suggesting a more complex scenario than a simple retreat to refugia and subsequent secondary contact. Notwithstanding their capability for long distance dispersal, effective gene flow between the reconnecting populations might have been low as seen in many contemporary wolf populations, including within the Iberian population [65–69]. Habitat changes caused by humans and persecution that culminated with the extirpation of central European wolves in the nineteenth century sealed the isolation of these populations even further.

An unexpected result of our analysis is the relatively old divergence time estimated between the Portuguese and Spanish wolves, that is similar in magnitude to the divergence between the Iberian and Italian populations, although with a large confidence interval: 6.2 (95% HPDI: 3–8.8) kya. Iberian wolves currently form a nearly continuous and expanding population in the North-western region of the Iberian Peninsula, spanning the Northern parts of both Portugal and Spain [15]. Our simulations indicate that G-PhoCS

would be capable of inferring correct estimates when using genomes sampled from a panmictic population or two very recently diverged populations. Since the relatively high inferred divergence time between Portuguese and Spanish wolves is unlikely to result from actual long-term geographical isolation between them, a possible alternative explanation is the existence of cryptic population structure in the Iberian wolf population. A recent study found that this population presents a high level of genetic structure, with up to 11 distinguishable subpopulations or clusters, that probably results from a general low dispersal of individual wolves [65]. Cryptic population structure has been described in several other apparently connected and uniform wolf populations and might reflect local adaptations to environmental and prey conditions [22,66–69]. The Portuguese and Spanish wolves used in the present study belong to distinct genetic clusters (Alto Minho and Castilla y León, respectively [9,65]), which might explain their differentiation. It is currently unclear if the population structure of the Iberian wolf population results mainly from the recent history of fragmentation and decline or reflects a long-term feature of the population, caused by ecological or geographical factors [65]. The inferred old divergence estimated in the current analysis might support the latter hypothesis.

Our G-PhoCS analysis also found evidence for post-divergence gene flow involving dogs and wolves, namely between Middle Eastern wolves and dogs, and from dogs to the Spanish wolf. The Israeli and Spanish wolf samples used in this study were previously found to have significant dog ancestry [8,9,42]. Post-domestication gene flow between dogs and wolves seems to have been common worldwide [9], especially in the Middle East, which also appears to be a hotspot for gene flow between different canid species [70]. The considerable gene flow from dogs into the Spanish wolf inferred by G-PhoCS is in accordance with the approximately 14–25% dog ancestry previously inferred for this sample [9,42]. Although hybridization between wolves and dogs in the Iberian Peninsula has been described in the extant population, wolves generally do not appear to present significant

levels of dog ancestry [71]. However, hybridization dynamics can be different at a local scale, as in some areas repeated hybridization and backcrossing have been detected [72]. This pattern might explain the difference in dog ancestry between the Portuguese and Spanish wolves, apparent in our study and that of [9].

Our study provides new insights into the historical processes that shaped the genetic diversity of Southern European wolves. The use of whole genomes corroborates the relatively recent origin of these populations (as well as that of all modern wolves) and supports their long isolation, possibly due to environmental changes at the end of the Pleistocene. This history might also explain the distinctiveness of the Italian and Iberian population seen at the genetic and morphological levels. Their low long-term effective population sizes, resulting from the historical bottlenecks and isolation, are also responsible for the standing genetic variation that these population have to face future environmental changes.

Data accessibility. This article has no additional data.

Authors' contributions. P.S., M.G., E.R., R.W. and R.G. designed the study; P.S., M.G., D.O.V. and Z.F. performed analyses; P.S., M.G., R.C., E.F., E.R., E.R., R.W. and R.G. contributed original data; P.S. drafted the manuscript. All authors critically revised the manuscript, gave final approval for publication and agree to be held accountable for the work performed therein.

Competing interests. The authors declare no competing interests.

Funding. P.S. was supported by PhD grant no. SFRH/BD/60549/2009 and under project PTDC/BIA-EVL/31902/2017 from the Portuguese Foundation for Science and Technology (FCT). D.O.V. was supported by a 2019 UC MEXUS-CONACYT collaborative grant and a DGAPA-PAPIIT grant (PAPIIT-IA200620). R.G. was supported by a research contract under DL57/2016 from FCT. This work was also partially supported by FCT (project PTDC/BIA-EVF/2460/2014 to R.G.) and by Norte Portugal Regional Operational Program (NORTE2020), under the PORTUGAL 2020 Partnership Agreement, through the European Regional Development Fund (ERDF) (NORTE-01-0145-FEDER-000007).

Acknowledgements. We thank the ECOGEN group at CIBIO for comments on an earlier version of the manuscript.

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
